# Association of CT-Derived Extracardiac Features and Aortic Annulus Size in Patients Planned for TAVI

**DOI:** 10.3390/jpm13020254

**Published:** 2023-01-30

**Authors:** Igor Volodarsky, Vladimir Perhulov, Galyna Gochman, Valeriu Cuciuc, Michael Welt, Gera Gandelman, Jacob George

**Affiliations:** 1Heart Center, Kaplan Medical Center, Hebrew University, Pasternak St., 1, Rehovot 76100, Israel; 2Imaging Department, Kaplan Medical Center, Pasternak St., 1, Rehovot 76100, Israel; 3Internal Medicine and Geriatric Department, Kaplan Medical Center, Pasternak St., 1, Rehovot 76100, Israel

**Keywords:** TAVI, aortic annulus, aortic stenosis, valve

## Abstract

Introduction: Transcatheter aortic valve implantation (TAVI) is commonly used to treat patients with stenosed aortic valves. CT is one of the crucial steps in planning TAVI to obtain measurements of the aortic annulus to choose an appropriately sized prosthesis. Incorrect measurements can lead to patient-prosthesis mismatch and other complications. However, some patients cannot undergo ECG-gated CT with radiocontrast because of the presence of radiopaque objects in the thorax, arrhythmia, renal failure, etc. Aim: To explore supplementary methods to improve aortic annulus sizing for TAVI by extracardiac measurements. Methods: We included all patients who underwent CT as part of TAVI planning. Measurements of femoral and iliac arteries and the femoral head cross-sectional area were performed. Results: CT scans of 139 patients were included in this study. Sixty-three patients (45%) were males. Mean age of the female patients was 79.6 ± 7.1 years and of the male patients was 81.3 ± 6.1 years. Mean aortic annulus perimeter among female patients was 74.3 ± 6 mm (range 61.9–88.2) and 83 ± 7.9 mm among male patients (range 70.1–74.3 mm). Mean diameters of common iliac, external iliac, and common femoral arteries were 9.2 ± 1.8, 7.6 ± 1, 7.6 ± 1 mm, respectively, for females and 10.2 ± 1.8, 8.5 ± 1.3, and 8.6 ± 1.4 mm for males. Mean perimeter of the femoral head (average value of right and left femoral heads) among the female patients was 137.8 ± 6.3 mm, and among male patients was 155 ± 9.6 mm. A significant correlation was observed between the perimeter of the aortic annulus and the perimeter of the femoral head (Pearson’s R^2^ = 0.224). The correlation between the aortic annulus perimeter and the femoral head perimeter was stronger among men than among women (Pearson’s R^2^ = 0.66 and 0.19, respectively). Conclusion: Femoral head diameter is associated with annulus size. This may help size the appropriate prosthesis in cases where the measurements by CT are in the border zone if corroborated by clinically driven data.

## 1. Introduction

TAVI is a widespread method commonly employed to replace a severely stenotic aortic valve. The technique consists of a transcatheter insertion of a prosthetic valve mounted on a nitinol expandable frame and deploying it in the aortic valve annulus, while the native diseased valve is obliterated by the radial force of the prosthetic valve. Several types of prosthetic valves for TAVI exist. Some of them are self-expandable when at body temperature with a restricting sheath removed, and some of them are mounted upon a balloon and expanded through balloon inflation. One of the most significant tasks prior to the procedure is to obtain correct measurements of the aortic annulus dimensions to choose a prosthesis of appropriate size [1,2]. Incorrect evaluation of aortic annulus size can lead to patient-prosthesis mismatch, paravalvular leak, and other complications, which adversely influence the patients’ outcome [3,4]. 

Currently, the most accepted way to determine aortic annulus size is with multidetector computed tomography (MDCT). When performed methodically and systematically, the procedure allows very precise sizing of the aortic annulus in most cases. However, there are patients in whom obtaining appropriate images is precluded by several conditions, such as the presence of radiopaque objects in proximity to the aortic valve causing shadows, motion artifacts due to failure to reach appropriate ECG gating, too small an amount of radiocontrast material, or failure to achieve proper timing of injection and image acquisition. Moreover, in many patients, the perimeter or valvular area fall within the border zone between two sizes of currently available prostheses, which makes sizing more challenging. 

The possibility to supplement the evaluation of aortic annulus dimensions indirectly, by estimations based on anthropometric measurements unrelated to the heart, hasn’t been addressed in the literature to our knowledge. We hypothesized that a correlation might exist between aortic annulus size and the size of several extracardiac structures, such as peripheral blood vessel size and the size of several skeletal structures. In this work, we examine whether it is feasible to facilitate choosing an appropriately sized prosthesis by measuring extracardiac structures without directly measuring aortic annulus dimensions. 

## 2. Methods

We retrospectively included in this study all patients planned for TAVI in the interventional unit of Kaplan Medical Center who had their MDCT scans of heart and peripheral vessels available on site. The MDCT was ECG gated and acquired using a high-definition 256-slice CT scanner in our center, Philips, iCT (Amsterdam, The Netherlands). The study was approved by the local institutional committee ((Institutional ethic board approval number KMC-0196-2020, approved 17/11/2020)).

### 2.1. CT Image Acquisition and Measurement

In brief, the CT protocol used a single-source multidetector gantry. Tube voltage was 80–120 V. First, a non-contrast ECG-triggered scan covering the region from the level of carina to 2 cm below the cardiac apex level was acquired. Then a contrast-enhanced scan was performed, covering the region from the level of the C7 vertebra down to the proximal one-third of the thighs. Generally, a contrast bolus of 60 mL was injected into the antecubital vein with a speed of 3 mL/s, followed by flushing with 30 mL of normal saline at 3 mL/min. Additional slight alterations in contrast delivery rate and volume were made by an experienced radiologist according to the patient’s body size, age, renal function, and left ventricular systolic function. Bolus tracking was used to determine the timing of image acquisition initiation. The tracker’s region of interest (ROI) was placed in descending aorta on the level of the aortic annulus, while the scan was triggered as soon as the image of the descending aorta on the ROI level reached a density of 140 H.U. or 100 H.U. above the native value. The reconstructed images had a spatial resolution of 0.6 mm within the plane and 0.4 through the plane and a temporal resolution of 83 msec.

### 2.2. Patient Selection

Between January 2016 and October 2020, 181 patients underwent MDCT as a part of TAVI procedure planning. We excluded patients with severe peripheral vascular disease, bicuspid aortic valve, prosthetic aortic valve, more than mild aortic regurgitation, and a history of hip joint replacement. MDCT scans of the remaining 139 were retrieved and reviewed by two blinded expert physicians. Severe peripheral vascular disease was defined as the narrowing of common and external iliac arteries and common femoral arteries by atherosclerotic disease resulting in a lumen diameter below 5 mm. The patients with risk factors for peripheral vascular disease, such as hypertension, diabetes, and dyslipidemia, couldn’t be excluded since they were expectedly widespread in patients with diagnosed aortic valve disease.

### 2.3. CT images Reconstruction and Measurements

The multiplanar images during the end-systolic phases were reconstructed with the help of a PACs workstation. For anatomical reconstruction and measurements, a Phillips portal was used. End-systolic phases were defined as time points at 30%, 35%, and 40% of the RR interval, while further choice between the three phases was left to the operator’s discretion.

Aortic annulus was defined as the level at which nadir points of the three aortic cusps could be seen in one plane. The Phillips portal was used to find the plane, which was thereafter checked and corrected manually as needed. Aortic annulus appears nearly elliptic in the annular plane. Following the reconstruction of the aortic valve annulus in the annular plane, the long and short axes of this ellipse were measured. Also, the perimeter and area of the aortic annulus were measured within the same plane (Figure 1 and Figure 2). 

Femoral head diameters and perimeters were measured upon a multiplanar reconstruction of the relevant anatomical region. The measurements were performed in at least two planes, all crossing each other in the center of the femoral head. The diameter measured in a plane lying in parallel with the acetabular rim plane was chosen (Figure 3). Patients with femoral joint replacement (even unilateral) or any deformation of the femoral head and acetabulum were excluded from the study. Patients within whom an even semicircular femoral head surface could be demonstrated (due to osteophytes or synovial changes) were excluded, too. 

Measurements of peripheral arteries were performed using curvilinear reconstruction. After the curvilinear reconstruction, multiple axial sections were chosen at the operator’s discretion on the levels of the common iliac artery, external iliac artery, and common femoral artery, provided that the section was free of any significant lumen-narrowing atherosclerotic plaques (Figure 4). 

All measurements were performed by two physicians (a radiologist and a cardiologist) independently. Interobserver variability was examined upon selected images.

### 2.4. Statistical Methods

The mean and SDs were presented. The two-tailed student’s test was used to determine the significance of differences regarding comorbidities between the groups of patients, and a Pearson’s R test was performed to determine the statistical significance of the correlation between anthropometric measurements and aortic annulus dimensions. *p* values < 0.05 were considered significant. Statistical analysis was performed using IBM SPSS version 21.0 (Armonk, NY, USA). 

## 3. Results

MDCT scans belonging to 139 patients were included in this study. Among the patients, 76 (55%) were females and 63 (45%) were males. Mean age of the female patients was 79.6 ± 7.1 years, and the mean age of the male patients was 81.3 ± 6.1. A total of 81 patients (58.3%) underwent TAVI with a self-expandable prosthetic valve and 58 patients (41.7%) with a balloon-expandable prosthetic valve. Since the main point of interest in this study lies within anthropometrics, we analyzed male and female patients separately for the two genders a priori considering different body proportions. The baseline characteristics of the patients are provided in Table 1. 

Mean aortic annulus perimeter among female patients was 74.3 ± 6 mm (range 61.9–88.2) and 83 ± 7.9 mm among male patients (range 70.1–74.3 mm). Mean diameters of common iliac, external iliac, and common femoral arteries were 9.2 ± 1.8, 7.6 ± 1, 7.6 ± 1, respectively, for females and 10.2 ± 1.8, 8.5 ± 1.3, and 8.6 ± 1.4 for males. As to the mean perimeter of the femoral head (average value of right and left femoral heads), mean ± S.D. among the female patients was 137.8 ± 6.3 mm (range 124.6–148.8, interquartile range 132–142), and among male patients was 155 ± 9.6 (range 132.9–175.3, interquartile range 150–160). 

There was a weak positive correlation between the perimeter of the aortic annulus and mean diameters of the common iliac, external iliac, and common femoral arteries (Pearson’s R^2^ = 0.08, 0.11, and 0.16, respectively, on the right side, and 0.14, 0.15, and 0.13, respectively, on the left side) with all p values significant (Figure 5).

A stronger correlation was observed between the perimeter of the aortic annulus and the mean (right and left) perimeter of femoral heads (Pearson’s R^2^ = 0.22). A good correlation was found as well between the area of the aortic annulus and the perimeter of the femoral head and also between the mean axis of the aortic annulus and the perimeter of the femoral head. When analyzing male and female participants separately, the correlation between aortic annulus area, perimeter, and mean axis in the gender-homogenic group versus mean femoral head perimeter was better among male patients than female patients (Pearson’s R^2^, respectively, 0.42, 0.66. 0.66 among males and 0.22, 0.19, 0.18 among females. The p values were significant with respect to all tested correlations. Provisional formulae to calculate aortic annulus area, perimeter, and mean axis using femoral head dimensions were derived as linear functions of a trend line built based on Pearson’s correlations. As good correlations were found only in male patients, we defined these formulae only for this group of patients. The provisional formulae, each beside its relevant diagram and trend line, are shown in Figure 6.

## 4. Discussion

Use of appropriate imaging and retrieving accurate measurements before TAVI is mandatory. Contrary to surgical aortic valve replacement, sizing the aortic valve cannot be done under direct vision [5]. The best measurements to represent the size of the aortic valve are taken at the aortic annulus plane [6]. The line at which aortic valve leaflets are inserted into the aortic wall has a crown shape with three prongs pointing cranially. Each prong tip represents a commissure of leaflets. Between each pair of prongs, there is a leaflet with a nadir (i.e., the point closest to LVOT) in each leaflet situated roughly halfway between each pair of prongs. A virtual line that connects three nadir points within a single plane is defined as the aortic annulus. The current guidelines recommend using the following parameters to evaluate aortic annulus size as measured upon MSCT: long axis length of the aortic annulus, short axis length, perimeter, and area. Mean diameter is calculated as an arithmetical mean of long and short axes lengths, while the diameter of the aortic annulus can also be derived from either perimeter or area. 

The annulus most frequently has an elliptic shape with slight eccentricity. It rarely is perfectly round, while the prosthetic valve is round to comply with the technical demand to expand with equal radial force in each direction following its deployment [6]. Regions of malapposition of the prosthesis to the aortic wall are sources of paravalvular leak, the worst prognostic factor following TAVI [7,8]. Although self-expandable prostheses, when compressed in one axis, usually expand in an orthogonal axis, thus assuming an elliptical shape, this feature has a limited impact [9]. More commonly, the aortic annulus assumes an almost rounded form, the same as the prosthetic valve, with the same perimeter as the aortic annulus used to have prior to implantation. Therefore, it is recommended to systematically implant a prosthesis slightly oversized with respect to the actual aortic annulus to force the annulus into a round shape and prevent the slit-like openings through which paravalvular leak becomes manifest [10]. 

When aortic valve prosthesis is significantly smaller than the actual aortic annulus size, it results in patient-prosthesis mismatch, hemodynamically expressed as residual aortic stenosis, and paravalvular leak, i.e., residual aortic regurgitation. Larger than expected aortic annuli are associated with a higher incidence of both residual paravalvular aortic regurgitation and residual aortic stenosis [11]. The situation when the prosthesis is too large for the patient is no less detrimental. In such cases, the prosthesis can apply more severe pressure on the conduction system and result in an atrioventricular block, requiring pacemaker implantation [12]. Coronary obstruction, improper prosthesis positioning requiring post-dilatation, or bailout valve-in-valve implantation can also stem from prosthesis-annulus mismatch [13,14,15,16]. Thus, a comprehensive toolbox is required to achieve optimal sizing with each available prosthesis. This is further emphasized due to the readouts that provide measurements that fall in the border zone of two prosthesis sizes.

The measurements must be performed upon images acquired with ECG gating, i.e., in a certain phase of the cardiac cycle, while images reconstructed using “mixed” phases are irrelevant for retrieving accurate measurements. However, in several patients, a correct ECG gating is hard to achieve due to an overly fast or irregular heart rhythm. Sometimes the amount of radiocontrast material in the heart chambers of interest is unsatisfactory because the timing of the scan following injection was evaluated erroneously due to the patient’s low cardiac output. Radiocontrast administration exposes patients to the risk of acute kidney injury, yet, attempts to diminish the amount of radiocontrast material to spare a patient’s renal function can also lead to suboptimal quality of the acquired images and failure to measure aortic annulus with required precision [17,18,19]. In such cases, transesophageal echocardiography (TEE) can replace MSCT, although it is less accurate [1]. Additionally, sizing of the valve can be performed during the procedure itself immediately before the TAVI by means of aortography. However, this modality is significantly less accurate than MSCT [1]. Consequently, there is a subpopulation of patients in which accurate measurements of the aortic annulus are difficult to achieve. In such cases, a score that affords a satisfactory evaluation of aortic annulus size without actually reconstructing its image, i.e., evaluating indirectly, would be useful. Before the advent of MSCT, sizing of the prosthesis was based on (TEE) and on clinical patient variables, such as body height and weight. 

We showed a good correlation between femoral head size and aortic annulus dimensions. The femoral head can be imaged without ECG gating, without radiocontrast material, and with a relatively small amount of radiation. As our results show, if no MDCT angiography had been performed and picking the appropriately sized prosthesis had been based exclusively upon skeletal dimensions, 60% of patients would still have received prosthetic valves of the same sizes as they did. We chose the femoral head diameter and perimeter among other skeletal structures because this structure has an almost regular spheroid shape whose boundary is readily definable and whose dimensions are easily measured. Moreover, it was available in all pre-TAVI MDCT scans. Several other structures, such as the femoral neck or clavicle length, were discarded because of their irregular shape (for example, the clavicle bends into alternating directions at angles that differ from patient to patient, as does the angle at which the femoral neck is affixed to the femoral shaft) or because not all of them were available on the scans. 

It is reasonable to anticipate that both the aortic annulus and femoral head correlate well with overall body dimensions. The femoral head is situated near the body’s center of gravity, almost exactly midway in a person’s stature, and bears more than half of body weight. The aortic annulus, in turn, an element of the fibrous skeleton of the heart, consists of more fibrous tissue and less elastic and muscular tissue, such as the vascular wall, elsewhere throughout the body. Therefore, among different measurements of the aorta in other planes, the aortic annulus is less prone to changes in size depending on loading conditions such as systemic arterial hypertension, valvular regurgitation, etc. A weaker correlation between peripheral arterial diameters and aortic annulus indirectly corroborates this statement. 

The association between the aortic annulus and the skeletal structure can be explained by causes beyond straightforward dependence of any organ dimensions on overall body size. The formation and maintenance of the fibrotic skeleton of the heart (of which the aortic annulus is a part) and the formation of bony structures have much in common on a genetic level [20]. To our knowledge, the most documented influential cytokines are the transforming growth factors (TGF) family, which includes TGF-β and bone morphogenetic proteins (BMPs) [21]. BMPs, through BMP receptors, act via the signaling SMAD proteins that participate in both heart formation, such as BMP-4 and BMP-10, and the formation of cartilage and bone, such as BMP-2 and BMP-3 [22]. SMAD proteins, involved in bone formation, have also been shown to participate in the embryonic development of heart valves and related structures. For instance, SMAD6 participates in the embryonic formation of the aortic valve, and its mutations have been found in some familial cases of the bicuspid aortic valve (BAV) [23,24]. The role of TGF-β and pathways generated by its action in the development of heart valves and skeletal structures have been thoroughly investigated. TGF-β signaling is implicated in several syndromes involving the heart valves and aorta and alterations in skeletal morphology, such as Marfan syndrome and Loetz-Dietz syndrome. For instance, type 2 Marfan syndrome and Loetz-Dietz syndrome, which comprise aorta and aortic annulus dilatation and aortic regurgitation, may be caused by mutations in TGF-β receptors [25,26,27,28,29]. Heart valve malformations are found in different genetic skeletal disorders involving gene coding for extracellular matrix structural proteins, such as collagen [30,31].

Intriguingly, the association between femoral head size and aortic annulus size was stronger among males than females. It’s hard to produce an educated explanation for this observation. Among the genetic mutations and related diseases mentioned above, one can raise a possibility that different activation of these genes in either gender can be responsible for the uneven incidence of heart disease phenotypes or morphological variants (such as a bicuspid aortic valve or mitral valve prolapse) in the two genders. It is known that a bicuspid aortic valve is 3–4 times more frequent in males than in females. On the other hand, mitral valve prolapse is more frequently encountered in women than in men. Both clinical conditions at least partially have genetic etiology. On the other hand, the different findings in the two sexes in our study can also be merely a statistical finding because body size among males was more variable (wider distribution of height and weight parameters and wider range of variability of femoral head size itself).

Additional skeletal structures unrelated to the cardiovascular system can be evaluated. We can speculate that if good correlations between aortic annulus dimensions and several skeletal structures’ dimensions are found, then a score with higher predictive value can be built, employing several structures. Structures that should be potentially considered include clavicle length, sternum length, and dimensions of bodies of vertebrae (but not their vertical dimensions, prone to changes due to osteoporosis). Prospective studies regarding patients’ outcomes following the implantation of prostheses chosen with the help of this hypothetical score may aid in assessing the significance of extracardiac measurements.

### Limitations

This is a hypothesis-generating retrospective study, and the sample size is relatively small. Further studies with larger patients from different populations are needed to corroborate our findings. Moreover, the predictive impact of the measurements and their potential added value over the mere assessment of the aortic annular perimeter/area should be tested in a prospective trial where prosthesis sizing prior to TAVI will be guided by these findings. We continue collecting data into the TAVI registry in our medical center. We will re-evaluate the data repetitively as the study population grows, and we look forward to publishing new results as soon as they appear.

The sizing of the prosthesis by extracardiac measurements has limited efficacy. It won’t be able to provide the height of coronary arteries ostia, which is also of great importance to prevent obstruction brought about by the bioprosthetic valve implantation. The decision regarding the size and type of the bioprosthesis still needs to be based on multiple imaging facilities, such as echocardiography, coronary angiography, and ECG-gated MDCT without radiocontrast material. It must also be kept in mind that estimating aortic annulus size by femoral head measurement should be used only in cases where conventional imaging is either technically impossible or produces unclear or ambiguous results.

## 5. Conclusions

The results of our study imply that in very infrequent cases when an assessment of aortic annulus upon MDCT is ambiguous or the annular area/perimeter achieved is in the border zone between prosthesis sizes, extracardiac measurements including femoral head performed upon MDCT without radiocontrast material can aid in choosing the appropriately sized valve. It should be emphasized that this method can be used only in addition to conventional measurement methods, and only in a small group of patients. Clinical trials that would test the outcome of TAVR patients with and without using femoral head assessment for supplementing conventional annular assessment in borderline-sized measurements are to be taken as proof of the validity of our findings. 

## Figures and Tables

**Figure 1 jpm-13-00254-f001:**
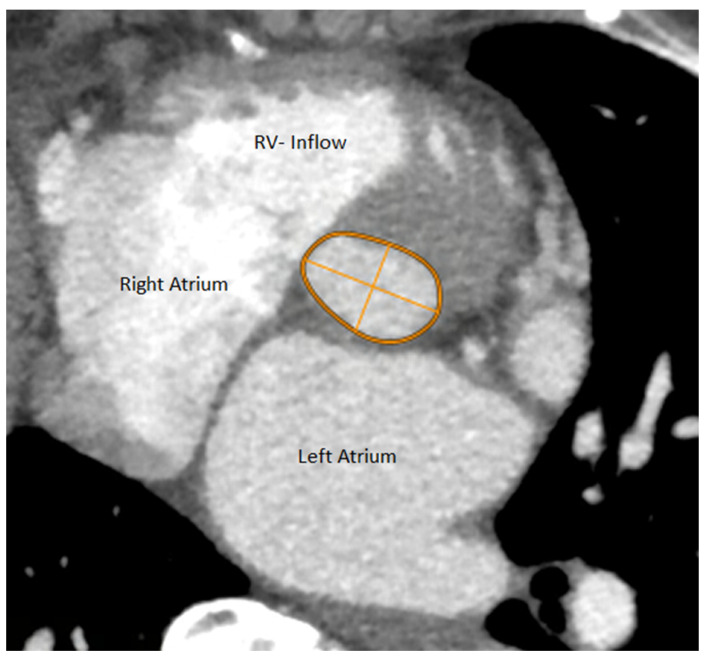
Slice through the heart within the Aortic Annulus plane with parameters measured upon MDCT.

**Figure 2 jpm-13-00254-f002:**
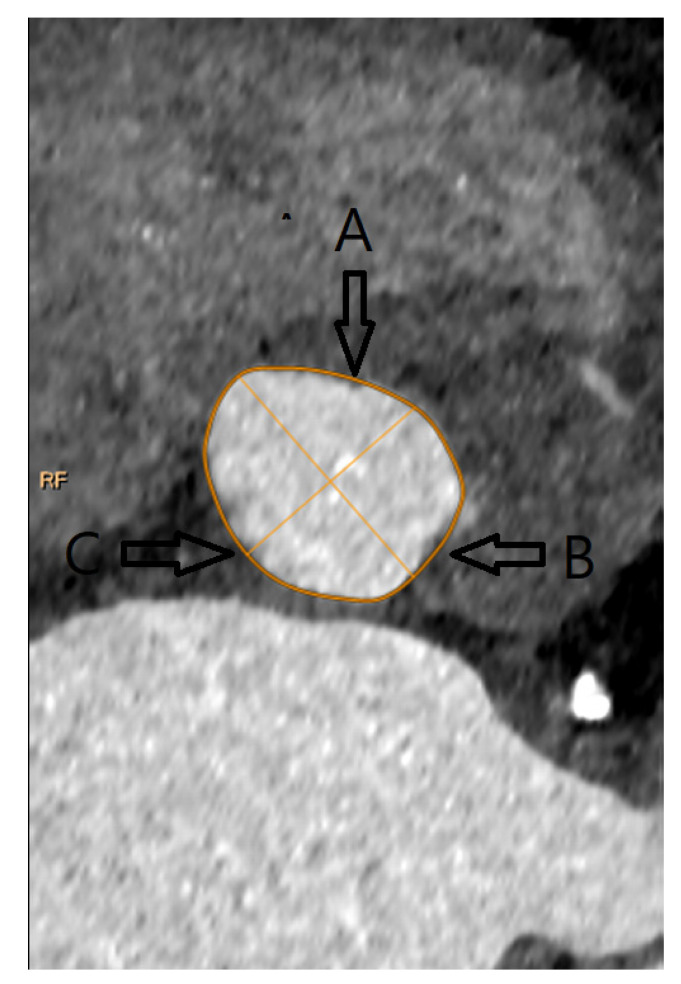
Aortic Annulus with main landmarks. (**A**): Right coronary cusp nadir; (**B**): Left coronary cusp nadir; (**C**): Non-coronary cusp nadir.

**Figure 3 jpm-13-00254-f003:**
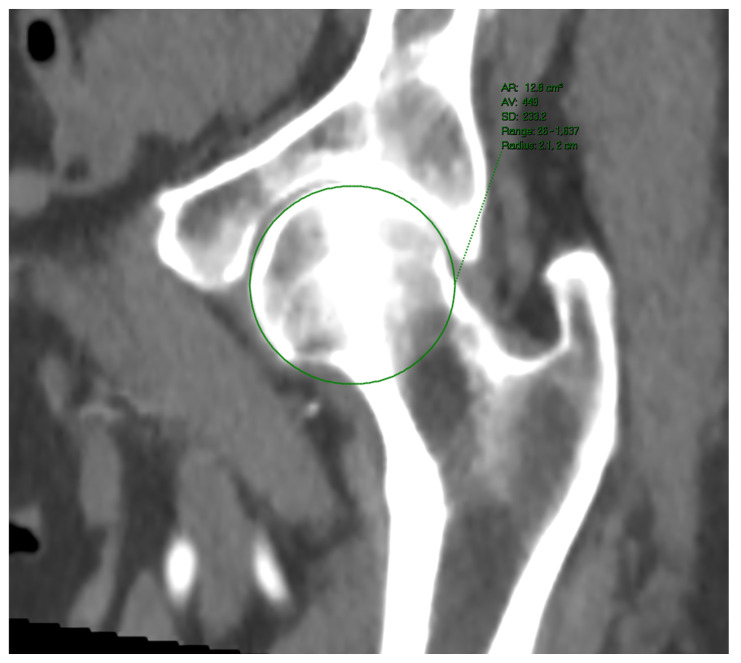
Femoral head perimeter measured upon MSCT.

**Figure 4 jpm-13-00254-f004:**
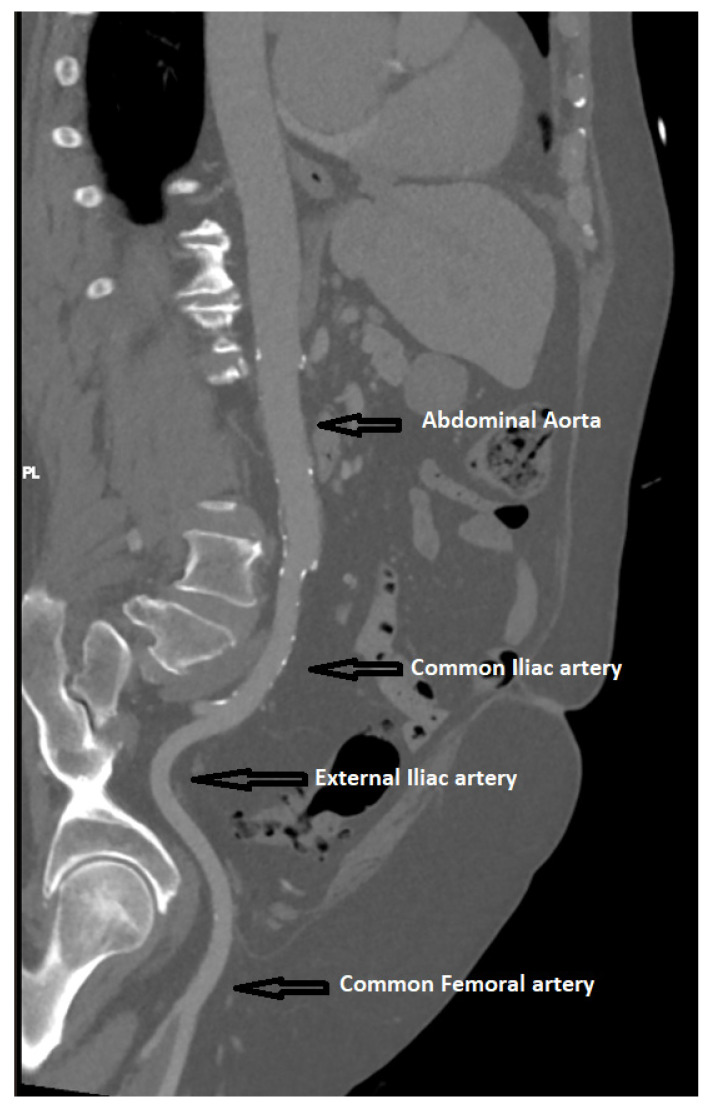
Measurement of peripheral arteries upon MDCT.

**Figure 5 jpm-13-00254-f005:**
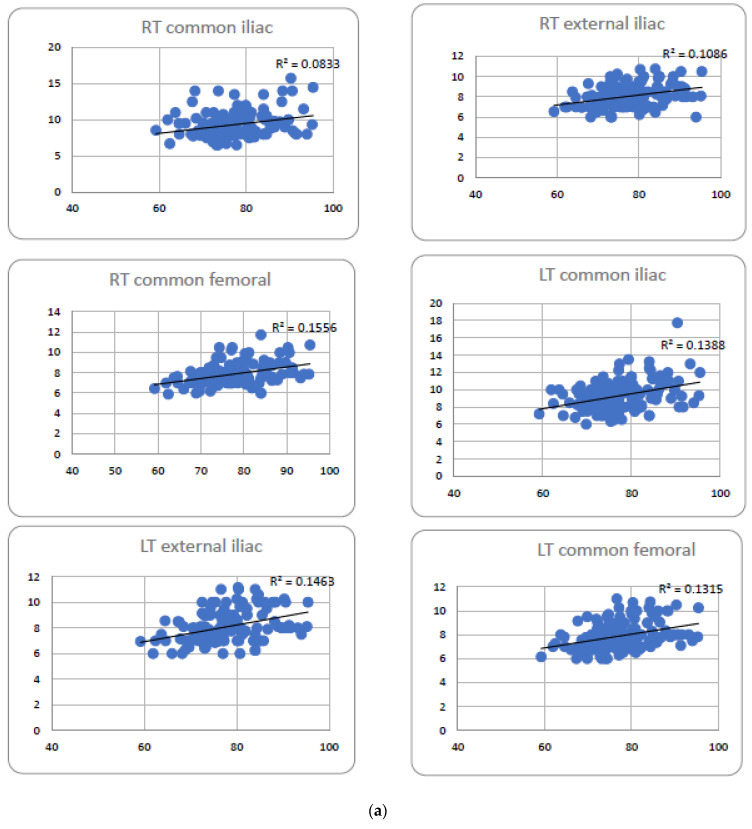
(**a**) Correlations of perimeters of the aortic annulus (mm, horizontal axis) versus peripheral vessels diameters (mm, vertical axis) among the total study population (*n* = 139); (**b**) Correlations of perimeters of the aortic annulus (mm, vertical axis) versus peripheral vessels diameters (mm, horizonal axis) male population (*n* = 63).

**Figure 6 jpm-13-00254-f006:**
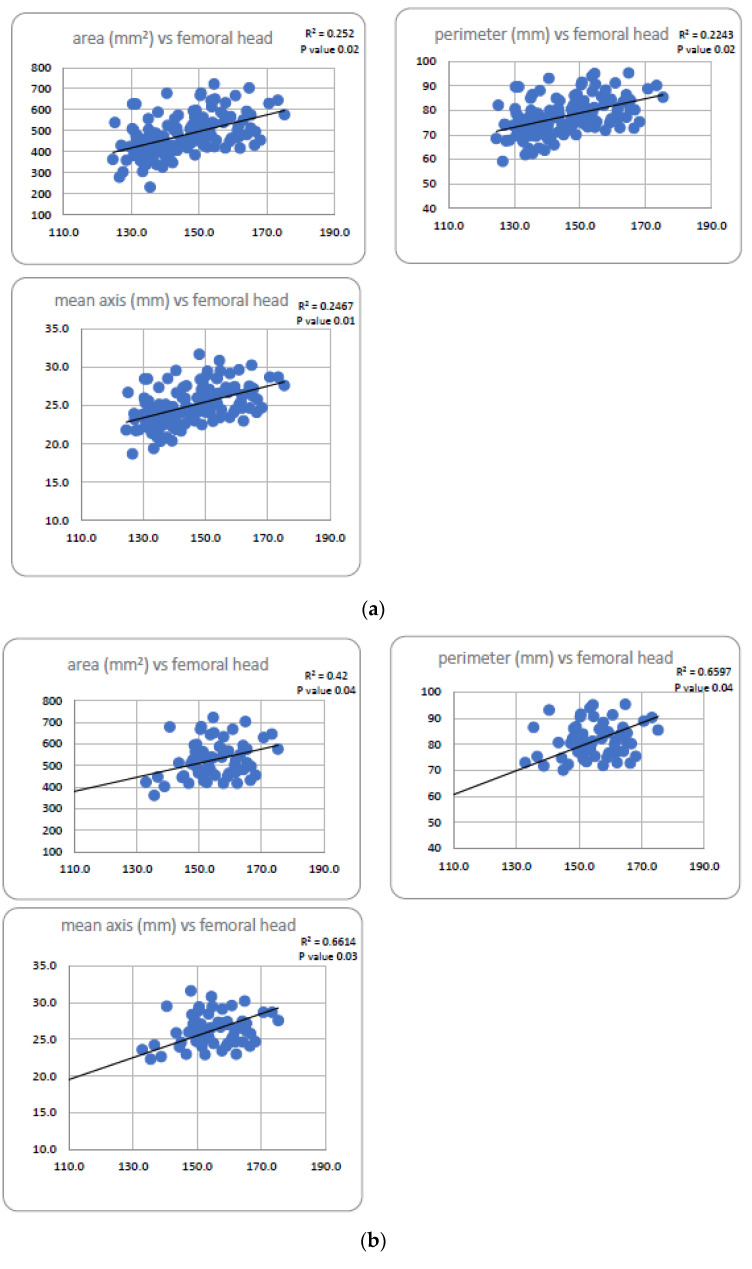
(**a**) Correlations of parameters of the aortic annulus (vertical axis) versus mean femoral head perimeter (in mm, horizontal axis) among the total study population (*n* = 139); (**b**) Correlations of parameters of the aortic annulus (vertical axis) versus mean femoral head perimeter (in mm, horizontal axis) among males (*n* = 63). Aortic Annulus area (mm^2^) = 3.26 * femoral head perimeter (mm) + 22.5; Aortic Annulus perimeter (mm) = 0.46 * femoral head perimeter (mm) + 10.2; Aortic Annulus mean axis (mm) = 0.15 * femoral head perimeter (mm) + 3; (**c**) Correlations of parameters of the aortic annulus (vertical axis) versus mean femoral head perimeter (in mm, horizontal axis) among females (*n* = 76).

**Table 1 jpm-13-00254-t001:** Baseline characteristics.

	Total (*n* = 139)	Male (*n* = 63)	Female (*n* = 76)
Age (years)	80.7 ± 7	81.3 ± 6.1	79.6 ± 7.1
Male gender (%)	63 (45)		
BMI	29.1 ± 5.6	28.6 ± 4.8	29.4 ± 6.7
BSA	1.78 ± 0.2	1.87 ± 0.2	1.71 ± 0.2
Hypertension (%)	114 (84)	55 (87)	59 (82)
Diabetes Melitus (%)	53 (39)	25 (40)	28 (38)
Current Smokers (%)	21 (15)	12 (19)	9 (12)
OCAD	67 (50)	27 (59)	23 (42)
PVD	32 (24)	10 (16)	9 (12)
status post-MI	19 (14)	11 (17)	6 (8)
Atrial Flutter or Fibrillation	40 (30)	18 (29)	21 (31)
s/p PCI	59 (44)	32 (51)	27 (38)
s/p CABG	17 (13)	10 (17)	8 (10)
s/p CVA	23 (17)	13 (27)	7 (9)
COPD	15 (13)	4 (7)	11 (14)
CKD 3-5	60 (43)	23 (37)	21 (31)
ACE-I or ARB	72 (63)	42 (67)	48 (67)
Beta-blockers	75 (66)	40 (63)	50 (69)
diuretics	57 (50)	26 (41)	36 (50)
CCB	25 (22)	19 (30)	20 (28)
Lipid-lowering drugs	88 (77)	55 (87)	54 (75)
Anti-aggregant	71 (66)	46 (73)	44 (61)
Anti-coagulants	36 (32)	16 (26)	23 (32)

Legend: BMI: basal metabolic index; BSA: body surface area; OCAD: obstructive coronary artery disease; PVD: peripheral vascular disease; PCI: percutaneous coronary intervention; CABG: coronary artery bypass grafting; CVA: cerebrovascular accident; COPD: chronic obstructive pulmonary disease; CKD: chronic kidney disease; ACE-I: angiotensin-converting enzyme inhibitors; ARB: angiotensin receptor blockers; CCB: calcium channel blockers.

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
