# Peer review of "Association of CT-Derived Extracardiac Features and Aortic Annulus Size in Patients Planned for TAVI"

_jpm, 2023, doi:10.3390/jpm13020254_

Round 1

Reviewer 1 Report

I would like to congratulate authors for their results in presented manuscript entitled: “ Association of CT-derived Extra-cardiac Features and Aortic Annulus Size in Patients Planned for TAVI”. They aimed to explore supplementary methods to improve aortic annulus sizing for TAVI using extracardiac measurements. I have major and minor comments on this:

 Major:

 1. To date in vast majority of cases aortic annulus measurements are based on echocardiography and CT. There is quite enough for planning the TAVI for experienced clinics. I think proposed theory in the manuscript is reserved for low or unexperienced teams in TAVI. However, to my mind, if the team has not enough experience they should not perform the operation like TAVI and, moreover, use additional criteria for the prosthesis implantation as a base. It may be harmful for the patient.

 2. Did authors took into account any skeleton pathology or bone pathology of the lower extremities? What to do in case of  osteophytes, thickening of the periosteum, etc.

 3. The number of patients is inadequate for the good statistical analysis.

Minor:

 1. The term ‘Transcutaneous’ in TAVI abbreviation is incorrect. Please replace it by ‘transcatheter’

 2.Authors should add the introduction short explanation of the aortic annulus-femoral head link.

 3. I think that the technique of the TAVI should be described, in short

 4. More details of the aortic annulus measurements are needed

 5. Two images in the manuscript has the same number (Figure 1). On the second of them the letters are poorly distinguishable.

 6. There is no any details of measurements of the common iliac, external iliac and common femoral arteries in the methods section.

 7. Please give us formulas origination. How did you come to this? more explanation is needed.

 8. Primarily authors used abbreviation MDCT then they has changed it to MSCT . Which is correct?

9. Table abbreviations deciphering is needed.

Reviewer 2 Report

Annular diameter measurement is one of the most important steps in the TAVI method for aortic stenosis.
I would like to congratulate you for doing a study on this subject.
I have read your work with great care. The strength of the study is that it includes so many cases from a single center.
However, patients with aortic stenosis may also have aortic regurgitation.
I believe that subgroup analysis should be performed especially in patients with aortic insufficiency.

Round 2

Reviewer 1 Report

No additional comments

Author Response

As I see that you have no additional comments, we use this opportunity to express our gratefulness for having contributed your time for revision of our paper.